# The SIGMA rat brain templates and atlases for multimodal MRI data analysis and visualization

D.A. Barrière [1,2,3,8], R. Magalhães[1,2,4,5,8], A. Novais[4,5], P. Marques[4,5], E. Selingue[1,2], F. Geffroy[1,2], F. Marques[4,5], J. Cerqueira [4,5], J.C. Sousa[4,5], F. Boumezbeur[1,2], M. Bottlaender [1,2], T.M. Jay[3], A. Cachia [3,6,7], N. Sousa [4,5]* & S. Mériaux [1,2]*

Preclinical imaging studies offer a unique access to the rat brain, allowing investigations that go beyond what is possible in human studies. Unfortunately, these techniques still suffer from a lack of dedicated and standardized neuroimaging tools, namely brain templates and descriptive atlases. Here, we present two rat brain MRI templates and their associated gray matter, white matter and cerebrospinal fluid probability maps, generated from ex vivo $T_2^*$-weighted images (90 μm isotropic resolution) and in vivo $T_2$-weighted images (150 μm isotropic resolution). In association with these templates, we also provide both anatomical and functional 3D brain atlases, respectively derived from the merging of the Waxholm and Tohoku atlases, and analysis of resting-state functional MRI data. Finally, we propose a complete set of preclinical MRI reference resources, compatible with common neuroimaging software, for the investigation of rat brain structures and functions.

---

[1] NeuroSpin, Institut des Sciences du Vivant Frédéric Joliot, Commissariat à l'Énergie Atomique et aux Énergies Alternatives, 91191 Gif-Sur-Yvette, France. [2] Université Paris-Saclay, 91191 Gif-Sur-Yvette, France. [3] Université de Paris, Institut de Psychiatrie et Neurosciences de Paris, INSERM, 75005 Paris, France. [4] Life And Health Sciences Research Institute (ICVS), School of Medicine, University of Minho, 4710-057 Braga, Portugal. [5] ICVS/3B's, PT Government Associate Laboratory, Braga, Guimarães, Portugal. [6] Université de Paris, Laboratoire de Psychologie du Développement et de l'Éducation de l'Enfant, CNRS, 75005 Paris, France. [7] Institut Universitaire de France, 75005 Paris, France. [8] These authors contributed equally: D.A. Barrière, R. Magalhães.
*email: njcsousa@med.uminho.pt; sebastien.meriaux@cea.fr

Magnetic Resonance Imaging (MRI) has assumed a central role in the investigation of the brain anatomy and function. This role can be attributed to, amongst others, its multimodal capabilities and the ability to study different tissues and their properties by acquiring different contrasts exhibiting high spatial resolution in a single scanning session. MRI studies typically involve the analysis of several subjects divided into experimental groups. In addition, each subject may undergo acquisitions using a variety of modalities which may span several scanning sessions. Because each brain has a unique volume and shape, and is uniquely positioned within the scanner, comprehensive analysis requires standardized anatomical templates and spaces, which enable the spatial normalization of the MRI data and the common mapping of experimental effects, allowing the aggregation of multiple studies. Furthermore, there is a need for standardized atlases that allow the labeling of findings with anatomically and functionally relevant Regions-Of-Interest (ROIs).

In humans, the Talairach space and template[1] and the Montréal Neurological Institute (MNI) template[2,3] are the most frequently used. That said, many different atlases and brain segmentation schemes have been created for specific purposes, emphasizing sub-cortical[4–8] and cortical[4,5,9,10] parcellations, white matter tracts[11–13], as well as maps created using functional information[14,15] (for review see refs. [16,17]). However, while this plethora of tools, methodologies, templates and atlases is available for human studies, a comparable array is either non-existent or is still incomplete for rat studies.

Some neuroimaging resources are available for the rat brain, but they have some technical limitations leading to usability constraints. The Waxholm Space atlas (http://www.nitrc.org/projects/whs-sd-atlas), built from ultra-high spatial resolution ex vivo acquisitions of a single Sprague-Dawley rat brain provides both $T_2^*$-weighted anatomical and Fractional Anisotropy templates, as well as an atlas which delineates 76 anatomical structures. There is a high emphasis on sub-cortical regions, especially the hippocampus[18,19]. On the other hand, the Tohoku University atlas (http://www.idac.tohoku.ac.jp/bir/en/db/rb)[20], based on lower spatial resolution data acquired in vivo from thirty Wistar rats, provides a $T_2$-weighted template, Gray and White Matter (GM and WM) and CerebroSpinal Fluid (CSF) tissue priors, and an atlas with 96 lateralized anatomical areas. There the focus is on cortical areas taken from the Paxinos-Watson rat brain atlas. An older alternative integrated onto the DPABI (http://rfmri.org/dpabi) analysis platform has been built from a dataset of ninety-seven Sprague Dawley rats acquired in vivo. It provides an anatomical $T_2$-weighted brain template associated with a set of brain tissue priors, and a co-registered version of the Paxinos-Watson rat brain atlas[21]. Each of these resources is characterized by significant limitations. The use of the Waxholm resources may be hindered by large file sizes (resulting from the very high spatial resolution) when conducting analyses with 4D datasets such as functional MRI, as well as from having been created from a single subject, thus leading to the incorporation of specific-subject/acquisition related anatomical details and artefacts. Both these factors create difficulties when performing normalization procedures with widely used software. Finally, the Waxholm template does not provide its own set of brain tissue priors. The Tohoku and DPABI resources do not provide full brain coverage and the lower spatial resolution may limit their use with high-resolution acquisitions. Additional resources such as rat brain atlases published as PDF images, web-based navigators or image stacks do exist, but are not compatible with the common neuroinformatic formats required by standard brain imaging software[22–24] (Table 1).

**Table 1 Comparison of rat brain templates currently available in the literature (OB Olfactory Bulb, HB HindBrain).**

| Name | Modality | Magnetic field intensity | Anatomical contrast | Spatial resolution of acquired images | Acquisition type | Number of animals | Rat strain | Template | Atlas | GM, WM and CSF priors | Brain coverage | MR image format | References |
|---|---|---|---|---|---|---|---|---|---|---|---|---|---|
| Waxholm | Anatomy/Diffusion | 7 Tesla | $T_2^*$ (GRE) | 39 × 39 × 39 µm³ | Ex vivo | 1 | Sprague-Dawley | Yes | Yes | No | Full | NIFTI | Papp et al.[18], Kjonigsen et al.[19] |
| Tohoku | Anatomy | 7 Tesla | $T_2$ (RARE) | 125 × 125 × 300 µm³ | In vivo | 30 | Wistar | Yes | Yes | Yes | OB missing | NIFTI | Valdes-Hernandez et al.[20] |
| Schwarz (DPABI) | Anatomy | 4.7 Tesla | $T_2$ (RARE) | 150 × 150 × 100 µm³ | In vivo | 97 | Sprague-Dawley | Yes | Yes | Yes | OB missing | NIFTI | Schwarz et al.[21] |
| Ratat1 | Anatomy | 14.1 Tesla | $T_2$ (SEMS) | 50 × 50 × 200 µm³ | Ex vivo | 9 | Long-Evans | Yes | Yes | No | Full | No (JPEG, PDF) | Wisner et al.[22] |
| Karolinska | Anatomy | 4.7 Tesla | $T_2$ (RARE) | 110 × 110 × 50 µm³ | In vivo | 5 | Sprague-Dawley | Yes | No | No | OB and HB missing | NIFTI | Schweinhardt et al. (2003)[48] |
| Figini | Anatomy | 7 Tesla | $T_2$ (SEMS) | 170 × 170 × 580 µm³ | In vivo | 10 | Sprague-Dawley | No | Yes | No | OB and HB missing | NIFTI | Figini et al. (2015)[49] |
| Paxinos & Watson | Anatomy/Diffusion | 7 Tesla | $T_2^*$ (GRE) | 25 × 25 × 50 µm³ | Ex vivo | 5 | Wistar | Yes | Yes | No | Full | No (JPEG, TIF, PDF) | Johnson et al.[23] |
| Development Atlas | Anatomy/Diffusion | 7 Tesla | $T_2^*$ (GRE) | 25 × 25 × 50 µm³ | Ex vivo | 45 | Wistar | Yes | Yes | No | Full | No (PDF) | Calabrese et al.[24] |

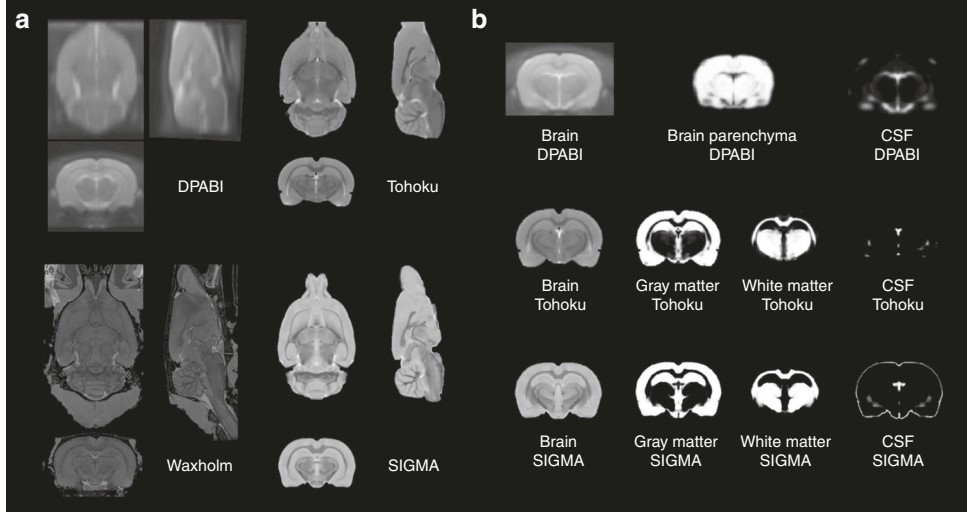

**Fig. 1 Comparison of anatomical templates of rat brain. a** Axial, sagittal and coronal views of the DPABI, Tohoku, Waxholm and SIGMA templates. **b** Coronal views of the tissue probability maps (Gray Matter, White Matter and CSF) associated with the DPABI, Tohoku and SIGMA templates. All the images have been linearly co-registered within the same space for comparison. For the DPABI template, only two tissue classes are available (brain parenchyma and CSF). Tissue probability maps are not provided with the Waxholm template.

**Table 2 Comparison of the Contrast-to-Noise Ratio (CNR) between the Gray Matter (GM) and the White Matter (WM) for the Waxholm, Tohoku and SIGMA anatomical templates of rat brain computed as proposed by Tapiovaara et al.[25]:**

$$CNR_{GM/WM} = \frac{|S_{gray} - S_{white}|}{\sqrt{\sigma_{gray}^2 + \sigma_{white}^2}}.$$

| Name | Number of animals | Spatial resolution of anatomical template | Number of voxels in gray matter | Mean signal in gray matter ($S_{gray}$) | Standard deviation in gray matter ($\sigma_{gray}$) | Number of voxels in white matter | Mean signal in white matter ($S_{white}$) | Standard deviation in white matter ($\sigma_{white}$) | Contrast-to-noise ratio between gray and white matter ($CNR_{GM/WM}$) |
|---|---|---|---|---|---|---|---|---|---|
| Waxholm | 1 | $39 \times 39 \times 39\ \mu m^3$ | 25,902,501 | 22,141 | 3265 | 12,165,815 | 18,981 | 3016 | 0.71 |
| Tohoku | 30 | $125 \times 125 \times 125\ \mu m^3$ | 642,257 | 24,299 | 3779 | 291,962 | 15,146 | 2605 | 1.99 |
| SIGMA | 6 | $90 \times 90 \times 90\ \mu m^3$ | 1,851,454 | 36,151 | 1627 | 979,977 | 26,019 | 4127 | 2.28 |

For the Tohoku and SIGMA templates, the GM and WM are defined from the corresponding tissue probability maps. For the Waxholm template, the GM and WM are defined from the labeled regions in the corresponding atlas. Source data are provided as a Source Data file

Herein, we present a new framework for rat brain imaging data analysis and visualization, one which combines a variety of MRI imaging types acquired in vivo and ex vivo, with resolutions suitable for both morphological and functional analyses. Combining the Waxholm and Tohoku atlases, we built a new anatomical atlas with a full brain coverage, which is complemented by a functional atlas derived from resting-state acquisitions data acquired from a large population of subjects. It was our goal to develop a comprehensive set of MRI references and resources, optimized for the rat brain, which would allow investigators to perform unified analyses of both structural and functional data. Use of these tools has been illustrated in the morphometric and connectivity analysis performed in this study.

## Results

**SIGMA anatomical template and tissue probability maps.** Using six ex vivo MGE acquisitions, we built a high-resolution (90 μm isometric voxel) anatomical template of the rat brain, covering the entirety of the organ, from cerebellum to olfactory bulb (Fig. 1a). When compared to other available resources (DPABI, Tohoku and Waxholm), only the Waxholm template provides similar coverage of the brain, although the contrast-to-noise ratio (CNR) is not optimal between gray (GM) and white (WM) matter structures. To further investigate this parameter for all resources, we computed the CNR between GM and WM for

the Waxholm, Tohoku and SIGMA templates as proposed by Tapiovaara et al.[25] (Table 2). We were able to confirm that our MGE imaging strategy, especially the parametrization of echoes and acquisition matrix, results in a high CNR between the two main tissue classes (WM and GM) and has a spatial resolution simultaneously capable of capturing anatomical detail while limiting the influence of magnetic susceptibility artifacts present in $T_2^*$-weighted images (Supplementary Fig. 1).

Tissue probability maps for both the SIGMA and Tohoku templates are shown in Fig. 1b. It is noteworthy that the Waxholm template corresponding to a single anatomical image does not provide any TPM, and the DPABI template defines only two of the three tissue classes (parenchyma and CSF) and as such is unable to distinguish GM and WM. Comparing the SIGMA and Tohoku templates and TPMs, several differences are evident through visual inspection (Fig. 2). They are: (i) more detailed definition of the WM fibbers of the corpus callosum (Fig. 2a); (ii) enhanced definition of the ventral hippocampus with finer delineation of GM and WM (Fig. 2b, c); (iii) more accurate delineation of the central and lateral ventricles in the CSF TPM (Fig. 2d).

**Template comparison for VBM analysis.** When we characterized the performance of the SIGMA and Tohoku templates during the VBM analysis of stress-induced change in the

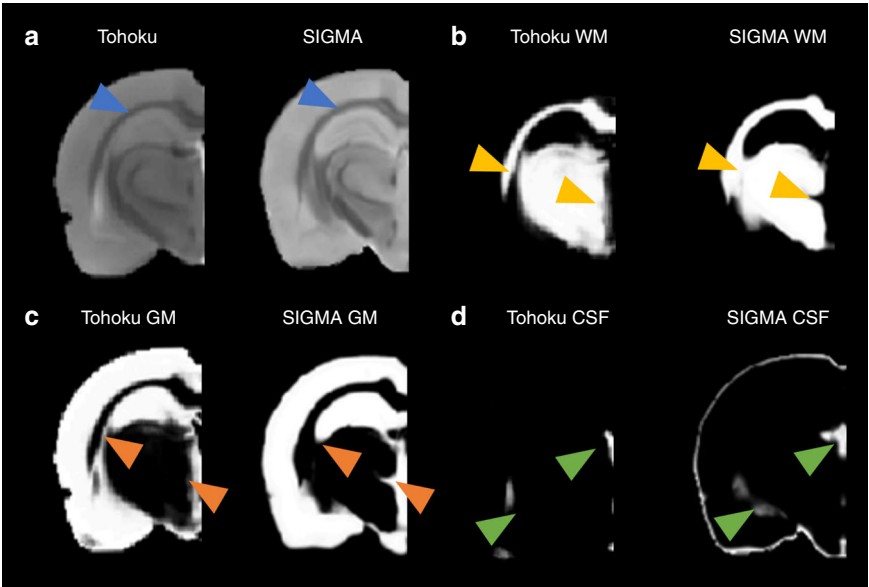

**Fig. 2 Comparison between Tohoku and SIGMA templates and tissue probability maps. a** Coronal views of the Tohoku (left) and SIGMA (right) anatomical template of rat brain at the same coordinates (Bregma: – 3.8 mm). Blue arrows indicate differences of corpus callosum shape between both templates. **b** Coronal views of the Tohoku (left) and SIGMA (right) White Matter (WM) probability maps at the same coordinates. Yellow arrows demonstrate differing classification of WM within the fimbria of the hippocampus. **c** Coronal views of the Tohoku (left) and SIGMA (right) Gray Matter (GM) probability maps at the same coordinates. Orange arrows show the differences in GM classification within the fimbria of the hippocampus. **d** Coronal views of the Tohoku (left) and SIGMA (right) Cerebrospinal Fluid (CSF) probability maps at the same coordinates. Green arrows show the differing CSF classification within the lateral ventricle.

hippocampus we found similar results. With both templates stressed animals demonstrated a global decrease of gray matter concentration (GMC) within the hippocampus in accordance with previously reported observations[26]. The observation was not dependent on the priors set used for the analysis, whether SIGMA (Fig. 3a, $t = 1.812$, df $= 10$, $p < 0.05$, 50 voxels, two sample Student $t$-test) or Tohoku (Fig. 3b, $t = 1.812$, df $= 10$, $p < 0.05$, 50 voxels, two sample Student $t$-test). We further explored this comparison between templates by building the distribution of $t$-values computed within the hippocampus. The histogram of these values reveals very similar distributions (Fig. 3c), with a higher kurtosis when performing the VBM analysis with the SIGMA template ($k_{SIGMA} = 3.29$, $k_{Tohoku} = 3.05$) and an accentuated skewness when using the Tohoku template ($sk_{SIGMA} = -0.34$, $sk_{Tohoku} = -0.52$). Furthermore, the VBM analysis with the SIGMA template reveals a larger number of voxels corresponding to $p < 0.005$ for the two sample Student $t$-test ($nvox_{SIGMA} = 878$, $nvox_{Tohoku} = 297$), as well as a wider range of $t$-values ($t_{SIGMA} = [-4.67–4.34]$, $t_{Tohoku} = [-4.06–3.08]$). Since the exposure to stress is expected to induce reduction in hippocampal volume[26], it would appear that the VBM analysis using the Tohoku priors gives a more conservative estimate of the effect, while a similar analysis using the SIGMA priors is more sensitive.

**SIGMA anatomical atlas of the rat brain.** Using the SIGMA anatomical template as a reference, we registered two MRI compatible atlases (Tohoku and Waxholm) of the rat brain into a common space. The process allowed us to overlap and merge the two atlases, creating a more comprehensive coverage of the rat brain (Fig. 4). From the Tohoku atlas, 124 labels (62 per hemisphere, cortical emphasis) were extracted, while from the Waxholm atlas, 122 labels (61 per hemisphere, subcortical emphasis) were used (Supplementary Table 1). The product of this synthesis was a new rat brain atlas comprised of 246 structures (123 per hemisphere). Comparing the volumes of the ROIs before and

after registration, we observed that the normalization procedure slightly increased the average volume of the regions of interest (+3.59%, Supplementary Table 2). For some structures, the volume variations could be accentuated due to the Voronoi diagram approach used to dilate competitively each ROI, and therefore fill the space within the brain mask to occupy the brain volume entirely. On the other hand, some white matter structures were found to be shrunk (i.e. fasciculus retroflexus), likely because the mask used to delineate the white matter tracts was calculated from the average of 6 animals, instead of just one as in the Waxholm atlas.

We visually inspected and carefully checked the results of the combination, and then reclassified and aggregated the labeled structures according to the systems to which they belonged (auditory, insular, temporal cortex, etc.), as well as with respect to anatomical topography (cortex, basal ganglia, etc.), tissue type (GM, WM and CSF) and hemisphere (left or right). Cortical structures were subdivided into functional (i.e. primary and secondary motor cortices) or structural (agranular, dysgranular, agranular/dysgranular, granular and posterior agranular insular cortices) areas according to the Paxions–Watson atlas (Fig. 5a, b). Sub-cortical structures such as the hippocampus (CA1, CA2, CA3, DG) and white matter tracts (optical tract, anterior commissure, SP5, etc.) were fully segmented, thus completing the labeling (Fig. 5c, d).

**Processing of fMRI data and functional atlas creation.** In order to identify noise and artifacts not accounted for during the standard processing of fMRI data, an Independent Component Analysis (ICA) was run for each scan, allowing the decomposition of the acquired signals into a multitude of temporal patterns. Each component of every scan was visually inspected and those identified as noise or artifact removed through a regression, with the residuals kept as signals of interest. The number of artifact components identified per scan ranged from 2 to 35, with an

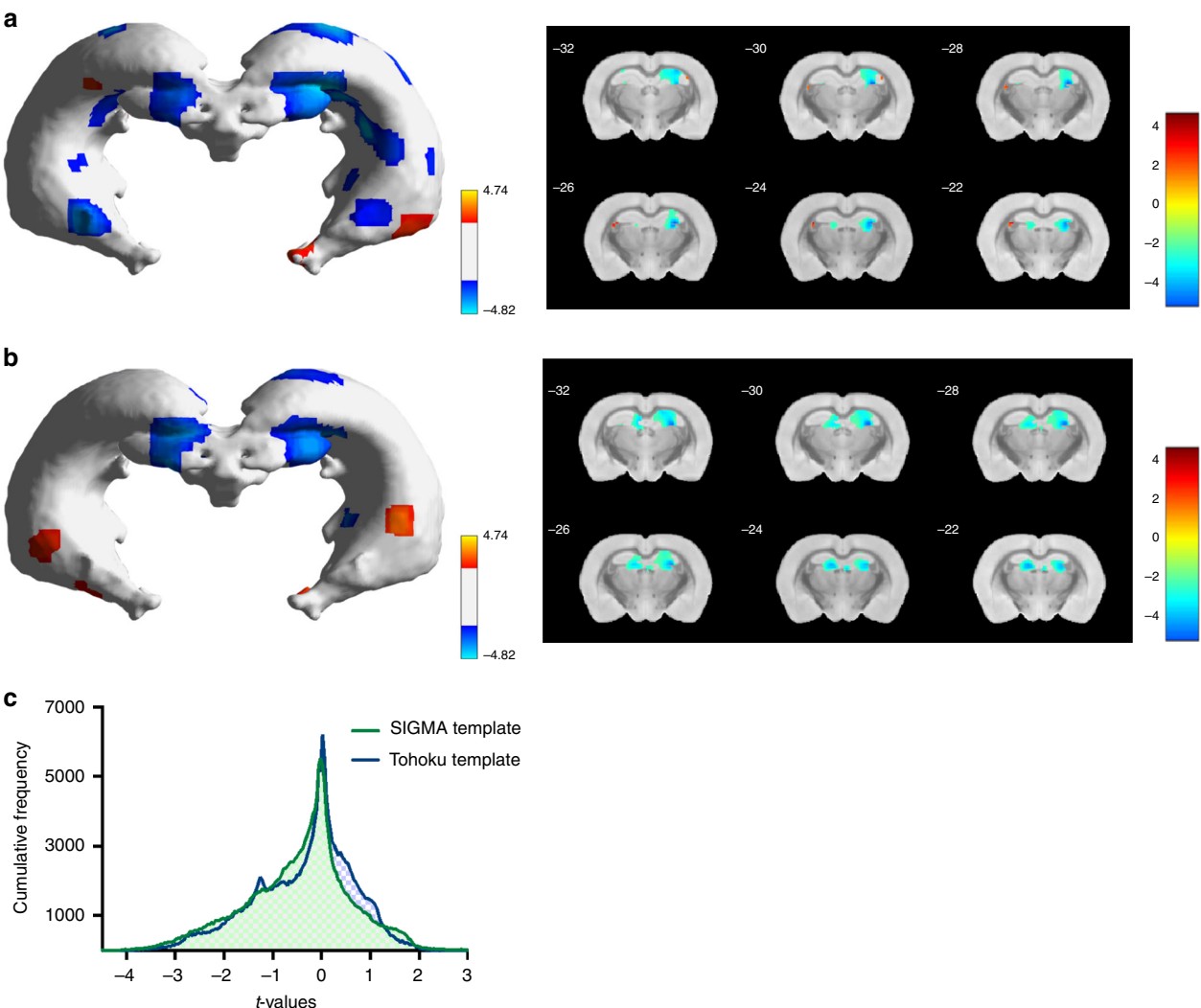

**Fig. 3 Comparison of VBM analyses performed using the SIGMA or Tohoku resources. a** Hippocampus mesh plot representing the surface map of Gray Matter Concentration changes between control and stress animals at the same threshold obtained from VBM analysis using the SIGMA priors and template. **b** Hippocampus mesh plot representing the surface map of Gray Matter Concentration changes between control and stress animals at the same threshold obtained from VBM analysis using the Tohoku priors and template. **c** Comparison of $t$-values distribution within the hippocampus when performing the VBM analysis with the SIGMA or the Tohoku template. Control and Stress animals ($n = 6$ for each) were compared using the two sample Student $t$-test implemented in SPM8 ($t = 1.812$, df $= 10$, $p < 0.05$, 50 voxels). Source data are provided as a Source Data file.

average and standard deviation of $15.2 \pm 5.7$, respectively. A number of sources of noise and artifacts were identified. One common artifact result from head movement, typically seen in voxel correlations all along the edge of the brain (Supplementary Fig. 2a). Another characteristic, which allows artifact components to be identified, is that they appear over a single coronal slice, while physiologically relevant signals would not be expected to be spatially restrained to one slice, acquisition-related events certainly could be. Artifacts of this type were found to exhibit very high $z$-scores in the equivalent space of one specific brain slice (or the equivalent after normalization and smoothing), resulting either in a partial saturation of the slice that resembles the shape of the brain (ghosting artefacts, Supplementary Fig. 2b, c), or in a complete saturation of the slice (antenna channel crosstalk artefact, Supplementary Fig. 2d).

Visual inspection of the 54 different components from the original group ICA resulted in none being identified as artifact. To test the reproducibility of the group ICA maps, we used a total number of permutations $N$ between 5 and 400 in steps of five.

From the variance analysis (Supplementary Fig. 3a), we observed stabilization after 280 permutations. When we ranked the components by reproducibility, we found stabilization with increasing $N$, with acceptable values after 100 permutations (Supplementary Fig. 3b, c). It was a significant finding that the ranking of the least reproducible components was very stable across all values, suggesting the choice of components excluded was stable also. At the conclusion of the analysis, we chose 300 permutations, as a safe stable point.

In the reproducibility analysis, we identified a logarithmic bimodal distribution similar to the one described in the original RAICAR methods[27], characterized by a local minimum at 0.63 (Supplementary Fig. 3d). All associated components were thresholded by this value and the reproducibility level was determined as the number of components above it. Fifty-three components were found to exhibit reproducibility above the half of maximum value of the reproducibility level (Supplementary Fig. 3e). Among the 53 reproducible components, 6 corresponded to bilateral symmetric ROIs. The result was the identification of

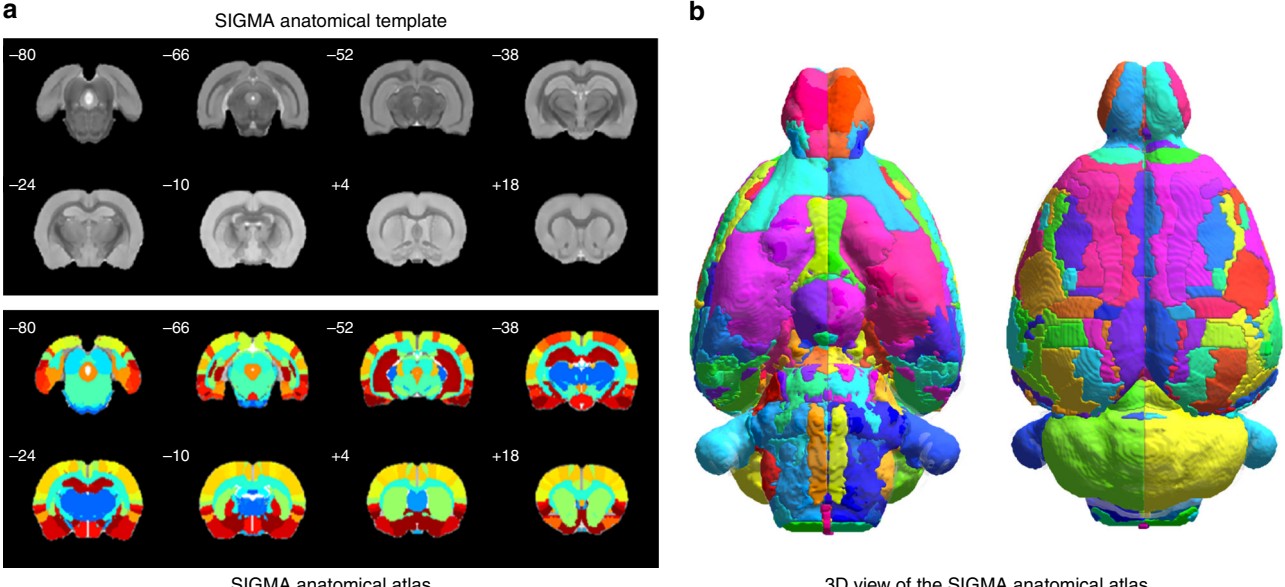

**Fig. 4 The SIGMA anatomical template and atlas of rat brain. a** Coronal slices of the ex vivo SIGMA anatomical template of the rat brain and the corresponding slices of the SIGMA anatomical atlas and (**b**) 3D representation of the SIGMA anatomical atlas.

59 distinct functional ROIs (labeled using the Paxinos–Watson rat brain atlas, Supplementary Table 3) that comprises the SIGMA functional atlas of the rat brain. The regions defined for the most part overlapped with anatomical cortical or subcortical structures, such as the hippocampus and the thalamus (Fig. 6).

**Comparison of structural and functional atlas networks**. We wished to assess the consequences of using our functional, as opposed to anatomical, atlas for the determination of functional connectivity. To do so, we compared the two during the analysis of rs-fMRI data. First, we tested the voxel-wise variance from each voxel matching a ROI, in order to evaluate how much each voxel diverges from the group mean. Comparing the voxel variance maps obtained with the functional and structural atlases, using non-parametric permutation tests, revealed significant effects in both directions for 91% of the voxels (both increased and reduced variance Fig. 7a). Increased variance was generally found using the anatomical atlas (59% of the voxels versus 31% for the functional atlas). The volumes of the ROIs of the reduced anatomical and functional atlases did not significantly differ ($t = 0.41$, df = 139, $p = 0.68$, hedge's $g = 0.074$, $x_{anatomical} = 16358 \pm 16210 \, \mu m^3$ and $x_{functional} = 17345 \pm 9984 \, \mu m^3$, confidence interval = [14455 19088], two sample Student $t$-test). The graph theoretical properties of both anatomical and functional connectomes were found to display a power-law distribution at the lower density values (between 0.025 and 0.1) and a transition to Gaussian distributions for higher values (Fig. 7b, c). Both networks were also found to have small-world like properties: average $\sigma_{anatomical} = 3.57$ ($c_a/c_{a\_rand} = 3.53$, $e_a/e_{a\_rand} = 0.98$) and average $\sigma_{functional} = 4.11$ ($c_f/c_{f\_rand} = 4.18$, $e_f/e_{f\_rand} = 1.02$) calculated at a scarcity of 0.1. When applying the Newman modularity algorithm to the connectomes, we found modularity scores of 0.86 and 0.84 (anatomical, functional), which resulted in partitioning the connectomes into 5 and 4 sub-modules (color-coded in Fig. 7b, c). For both connectomes, a grouping of neighboring anatomical regions emerged which was composed by a ventral network (colored in yellow), an anterior dorsal cortical network (red), a posterior cortical network in the functional atlas (teal) and a brainstem and posterior sub-cortical network (blue).

## Discussion

The use of preclinical MRI is one target of growing interest in the study of the brain structure and function in both healthy and pathological conditions. The use of advanced MRI techniques, coupled with the development of specific animal models, is a powerful way to push new breakthroughs in the understanding of brain functioning and pathology. Herein, our goal was to develop a comprehensive set of MRI compatible templates and atlases for the rat brain, to allow the unified and standardized analysis of multimodal rat brain MRI data and to pave the way for the development of multicentric preclinical studies (Supplementary Fig. 4).

We have described a new set of resources for the analysis of anatomical and functional MRI data acquired from the brain of the Wistar rat. The resources function in a common reference space and are: (i) a high-resolution ex vivo $T_2^*$-weighted anatomical template and its associated GM, WM and CSF tissue probability maps (TPMs) for voxel-based morphometry analyses, (ii) an in vivo $T_2$-weighted anatomical template and its associated GM, WM and CSF TPMs, (iii) an anatomical whole-brain atlas composed of 246 structures, derived from the merger of the Waxholm and Tohoku atlases and (iv) a functional brain atlas composed of 59 structures. In addition, we describe a brain mesh suitable for data visualization with common neuroimaging software (such as SPM, FSL, BrainVisa/Anatomist).

Using an ex vivo imaging approach based upon high spatial resolution MGE acquisitions, we were able to achieve high contrast between the GM, WM, and CSF in the rat brain samples and a high level of anatomical detail. Our protocol avoided the deformation which would have accompanied brain extraction from the skull. The parametrization of echoes ensured a high signal-to-noise ratio, while simultaneously restricting the influence of macroscopic magnetic susceptibility artefacts present in $T_2^*$-weighted images that are known to reduce the MRI signal within the entorhinal cortices and the amygdala. To these images, we applied the unified segmentation approach of SPM, which relies on a generative model combining bias correction with deformable tissue priors, and a Gaussian mixture model, to generate an anatomical template of the rat brain and corresponding GM, WM and CSF TPMs. The segmentation process

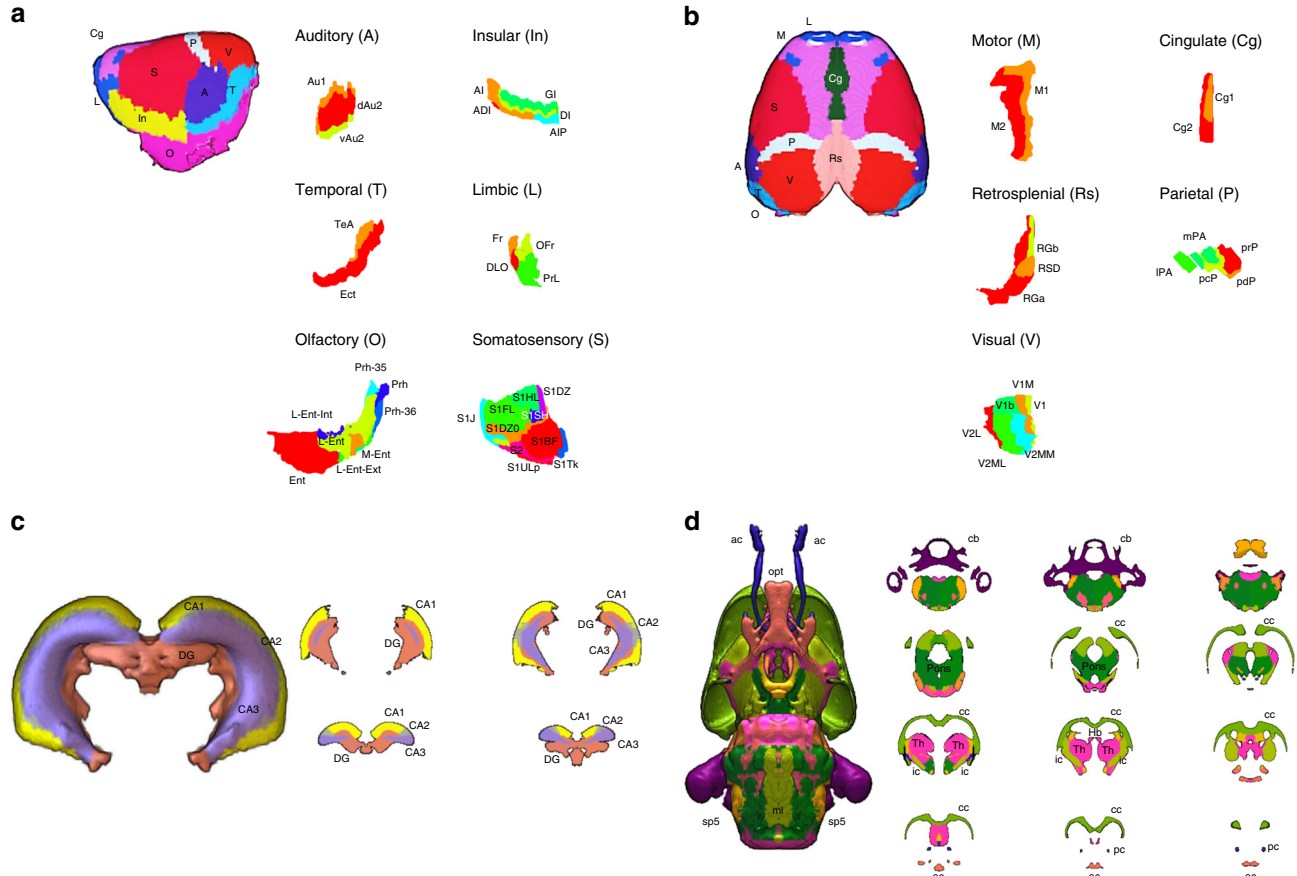

**Fig. 5 Cortical and sub-cortical details of the SIGMA anatomical atlas. a, b** Lateral and dorsal views of the cortical areas after normalization of the Tohoku atlas to the SIGMA anatomical template. The cortex has been segmented into cortical areas such as auditory (A), insular (In), temporal (T), limbic (L), olfactory (O), somatosensory (S), motor (M), cingulate (Cg), retrosplenial (Rs), parietal (P) and visual (V). Each area has been subdivided (using the Paxinos-Watson atlas) into functional areas (i.e. primary and secondary motor cortices) or structural areas (i.e. agranular, dysgranular, agranular/dysgranular, granular and posterior agranular insular cortices). **c, d** Lateral and dorsal views of sub-cortical structures (hippocampus and white matter tracts) after normalization of the Waxholm atlas on the SIGMA anatomical template. Legend of labeled regions:— Auditory: Au1 = primary auditory cortex; dAu2, vAu2 = secondary auditory cortex, dorsal and ventral areas. — Insular: AI, ADI = agranular insular and dysgranular insular cortices; GI, DI = granular and dysgranular insular cortices; AIP = posterior agralunar insular cortex. — Temporal: TeA = temporal association cortex; Ect = ectorhinal cortex. — Limbic: Fr = frontal association cortex; DLO = dorsolateral orbital cortex; OFr = orbitofrontal region; PrL = prelimbic cortex. — Olfactory: Prh = perirhinal cortex; Prh-35, Prh-36 = perirhinal areas 35 and 36; Ent = entorhinal cortex; L-Ent, M-Ent = lateral and medial entorhinal cortices; L-Ent-Int, L-Ent-Ext = lateral entorhinal cortex, internal and external parts. — Somatosensory: S1J, S1FL, S1HL, S1SH, S1ULp, S1Tk, S1BF = primary somatosensory cortex, jaw, forelimb, hindlimb, shoulder, upper lip, trunk and barrel field regions; S1DZ, S1DZ0 = primary somatosensory cortex, dysgranular region and dysgranular zone 0; S2 = secondary somatosensory cortex. — Motor: M1, M2 = primary and secondary motor cortices. — Cingulate: Cg1, Cg2 = primary and secondary cingular cortices. — Retrosplenial: RGa, RGb = retrosplenial granular A and B cortices; RSD = retrosplenial dysgranular cortex. — Parietal: lPA, mPA = lateral and medial parietal associative cortices; pcP, pdP, prP = parietal cortex postero-caudal, dorsal and rostral parts. — Visual: V1 = primary visual cortex; V1b, V1M = primary visual cortex, binocular and monocular areas; V2L = secondary visual cortex, lateral area; V2ML, V2MM = secondary visual cortex, mediolateral and mediomedial areas. — Hippocampus: CA1, CA2, CA3 = cornu ammonis areas; DG = dentate gyrus. — White matter tracts: ac = anterior commissure; cc = corpus callosum; f = fornix; Hb = habenular commissure; ic = internal capsule; ml = medial lemniscus; opt = optic tract; pc = posterior commissure; sp5 = spinal trigeminal 5 tract; Th = thalamus.

started with the existing Tohoku priors of GM, WM, and CSF, and the output TPMs were then used in an iterative workflow as new input priors. This process allowed significant improvement in the quality of the generated TPMs, as well as guaranteeing a complete coverage of the brain.

Compared to other templates available in the literature (DPABI, Waxholm and Tohoku), the SIGMA anatomical template exhibits a superior CNR between the two main tissue classes (GM and WM), accompanied by one of the highest spatial resolutions (90 μm isotropic). Only the Waxholm template has been acquired with a higher spatial resolution (39 μm isotropic). Therefore, we believe our template offers a good compromise between resolution and CNR and further, that our template with

associated TPMs will be of great utility for neuroimaging studies requiring both spatial registration and segmentation of MRI anatomical data. The creation of a second anatomical template, in the same space as the ex vivo SIGMA template but built using in vivo $T_2$-weighted acquisitions provides another resource for the registration of in vivo anatomical and functional MRI data such as is required with BOLD sensitive EPI acquisitions. We recognize and point out a peculiarity of our TPMs, which is that certain regions of sub-cortical gray matter (thalamus, PAG) are included in the white matter prior. We believe this is due to a high fibber density within these structures which results in intensity range similar to that of white matter. This is a common phenomenon, one also found in other standard resources (Fig. 1).

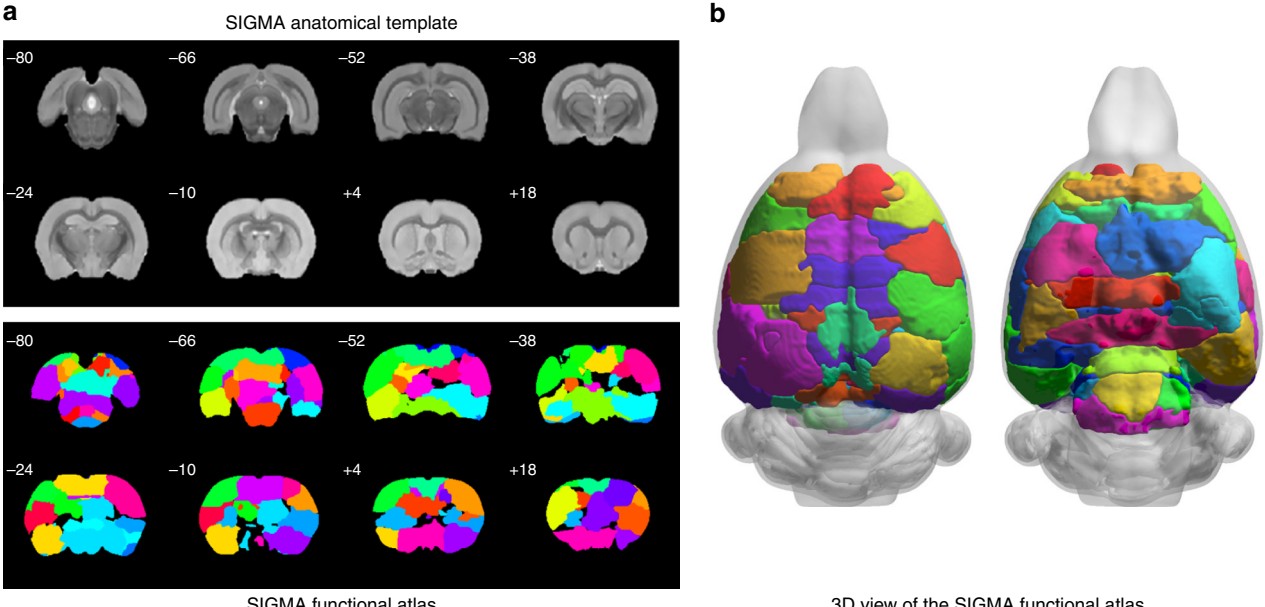

**Fig. 6 The SIGMA functional atlas of rat brain. a** Coronal slices of the ex vivo SIGMA anatomical template of the rat brain and the corresponding slices of the SIGMA functional atlas and **b** 3D representation of the SIGMA functional atlas.

To confirm the potential of the SIGMA anatomical template of the rat brain and its corresponding tissue probability maps, we tested their use as a registration reference by analyzing the MRI data acquired on a cohort of animals previously characterized from a behavioral (elevated plus maze) and neuro-endocrine (corticosterone assay) points-of-view[26]. When we compared the results obtained with the Tohoku and SIGMA templates using the same anatomical MRI dataset and the same SPM data analysis pipeline, similar findings in the expected direction were obtained. It is of interest that the use of SIGMA resources resulted in more prominent effects. Recent studies suggest that interpolation introduced by nonlinear normalization procedures promotes over-regularization. To address these issues, alternative methodologies such as minimal deformation templates have been employed in both human and primates studies[28–30]. It would thus appear reasonable to investigate using these alternative-processing strategies in rats to improve spatial normalization, particularly in longitudinal studies, by expanding the available resources to include age and strain-specific templates.

The creation of a complete MRI compatible atlas is a complex challenge considering the large number of structures and sub-divisions identified within the Paxinos–Watson reference atlas. The absence of gyri in the rat brain makes the parcellation of the cortical regions very challenging. Only through accurate registration of a MRI anatomical template with the Paxinos–Watson atlas, is such a detailed delineation feasible[18,20,21]. Currently there exist two MRI templates with corresponding atlas for the rat brain: the Waxholm atlas[18], which contains primarily sub-cortical structures, and the Tohoku atlas[20], which emphasizes cortical anatomy. While the two are complementary in their coverage, their differing spatial orientation and resolution make their use in the same study impractical or difficult. To address this issue, we registered these two atlases into the space of our SIGMA anatomical template. By so doing we are able to provide a comprehensive and detailed atlas of the rat brain for MRI morphometry and segmentation. To enhance its accuracy, we complemented this Waxholm–Tohoku merged atlas with an improved more accurate delimitation of the white matter and ventricles. We recognize that the segmentation of some sub-cortical regions (e.g. thalamus, amygdala) will need to be improved to provide a more detailed rat brain atlas in stereotaxic coordinates dedicated to MRI analysis.

To complement the anatomical atlas herein described, we developed a functional atlas for the rat brain, using a group ICA analysis validated through a RAICAR approach. From this analysis, we identified 59 bilateral ROIs covering cortical, sub-cortical and brainstem structures that are functionally distinct. Despite having been derived from purely functional data, this atlas broadly, if not precisely, correlates with the general anatomical boundaries and many are associated with specific anatomical structures. A primary motivation for the creation of this atlas is derived from the need to perform brain segmentations which is optimized for functional MRI analysis, since the signal sources do not necessarily match typical anatomical boundaries. A similar requirement has been identified by those performing human studies, resulting in efforts to generate more diverse, multi-modal atlases[14,15]. A significant result of our study is that by comparing voxel-ROI variance we have shown that the use of a functional atlas reduces variability, thus decreasing the loss of information resulting from the reduction of dimensionality. The present work was further motivated by the recognition of the need to develop resources having resolution parameters and ROI sizes appropriate for rat fMRI studies. Many anatomical atlases, such as the Waxholm or Tohoku atlases, are characterized by a large number of regions defined with a high spatial resolution. However, standard functional MRI acquisitions are unable to reach such high spatial resolutions while preserving an appropriate contrast-to-noise ratio or repetition time. As a consequence, either an up-sampling of the functional MRI datasets or a down-sampling of the atlas resolution is necessary when processing the data. Both these scenarios present drawbacks, so having appropriately designed and optimized resources can greatly improve and facilitate the analysis of functional MRI data. Using our rs-fMRI dataset, we validated the use of both our atlases for functional connectivity analysis by building whole brain connectomes. We demonstrated not only that the small-world metrics, degree distribution and nodal modularity for both networks fit the expected parameters[31], but also are comparable for both approaches. Users

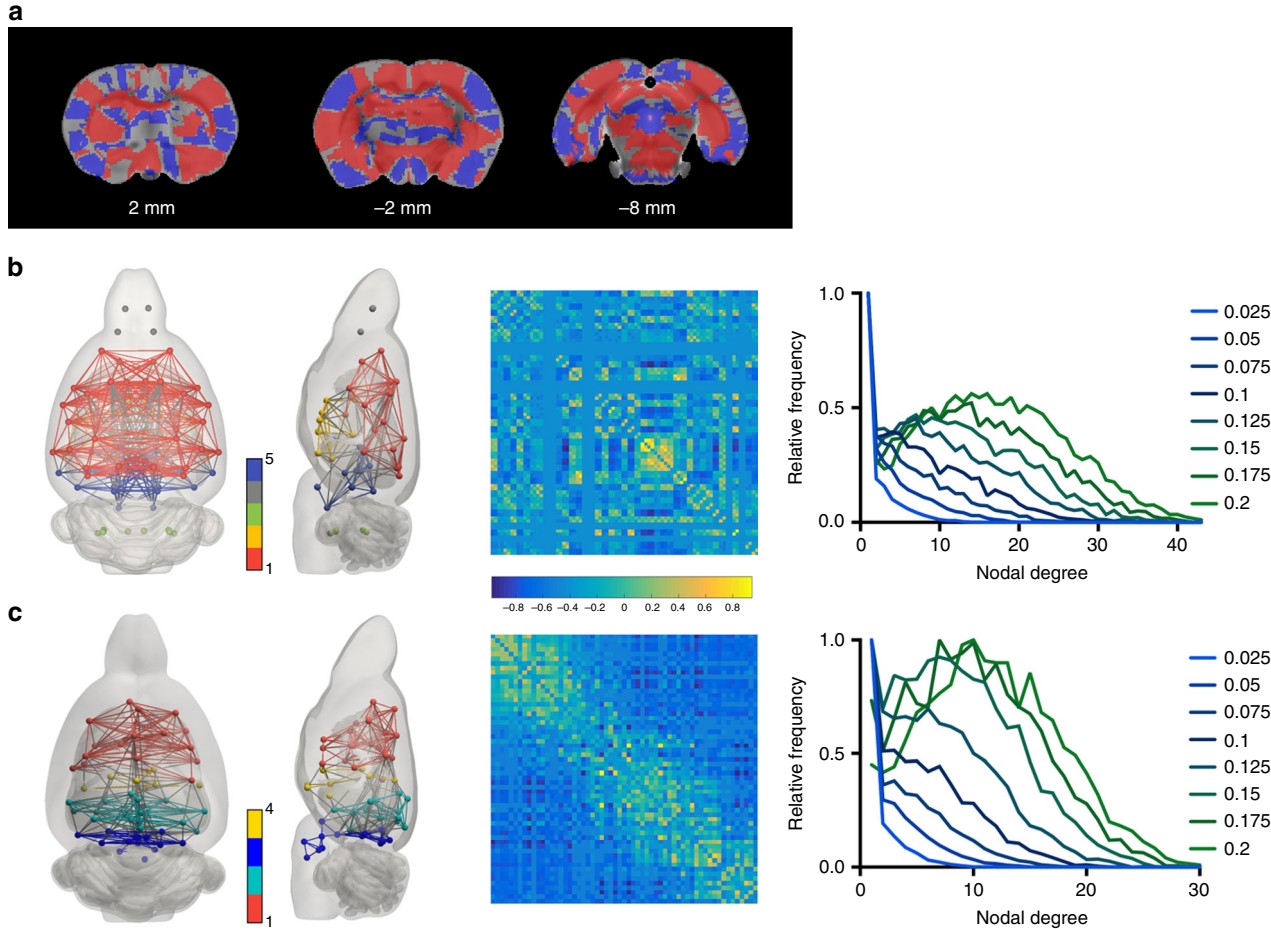

**Fig. 7 Functional connectivity analysis performed with anatomical or functional atlas. a** Significant differences ($p < 0.05$, non-parametric permutation-based group comparison with a Student $t$-test, corrected for multiple comparisons using Family Wise Error) in the voxel-mean ROI signal variability. Red represents reduced variability using the functional atlas while blue represents increased variability. Functional connectomes built with **b** the SIGMA anatomical atlas and **c** the SIGMA functional atlas. Each module within the network is coded by the node color. Shown here are the average networks visualized using BrainNet Viewer (left), the average connectivity matrices (center) and the plots of the degree distribution for densities between 0.025 and 0.2. ($t = 0.41$, df = 78, $p = 0.68$, two sample Student $t$-test). Source data are provided as a Source Data file.

thus have the flexibility to choose and adapt the unique requirements of their investigations. Moreover, the availability of highly consistent anatomical and functional atlases within the same reference space should allow their concomitant use and encourage testing of more complex hypotheses.

As a limitation, the SIGMA functional atlas has been built from resting-state fMRI acquisitions performed on anesthetized animals, using adapted concentrations of isoflurane permitting to keep animals in the same physiological conditions of breathing and temperature. Even if isoflurane is the most popular anesthetic product used for preclinical MRI studies[32], it is also known to obscure the naturally occurring functional connectivity by inducing synchronous cortico-striatal fluctuations and silencing the subcortical activity[33–36]. However, a prolonged fMRI acquisition on awake animals is still technically challenging. Therefore, we chose to rely on the most commonly used protocol for resting-state fMRI acquisitions to emulate a rat brain functional atlas that could be used to study functional connectivity alterations in isoflurane anaesthetized rat.

Animal models deliver crucial information for the understanding of brain structure and function both in healthy and pathological conditions. The SIGMA template and rat brain atlases were created to bridge the gap between the basic and clinical neurosciences by providing to the preclinical neuroimaging community specific resources built to be used in conjunction with the neuroinformatic tools and methodologies commonly used in human MRI studies. It is our hope that these resources will help basic neuroscientists to conduct their analyses of anatomical and functional datasets in a more standardized way, with the goal of reaching more reproducible conclusions.

## Methods

**Animals**. In total 47 male Wistar rats (Janvier, Le Genest-St-Isle, France) 8 weeks of age were housed in pairs and maintained on a 12-h light/dark cycle, at 55% humidity with access to food and water ad libitum. All in vivo experiments were conducted in strict accordance with the recommendations of the European Community (2010/63/EU) and the French legislation (decree no. 2013–118) for use and care of laboratory animals. Experimental protocols (stress protocol and MRI scanning) were approved by the "Comité d'Éthique en Expérimentation Animale du Commissariat à l'Énergie Atomique et aux Énergies Alternatives—Direction des Sciences du Vivant Ile-de-France" (CETEA/CEA/DSV IdF, protocol number ID 13-023).

All experiments and resources described in this work were executed and created as a part of the SIGMA project, a project co-financed by the Portuguese FCT and French ANR. The projects main goal was to study the impact of stress upon the rat brain revealing imaging biomarkers of resistance and susceptibility to stress. All the data herein presented were acquired on control animals. In the Results section, we describe two examples of the application of the SIGMA resources on the analysis of data resulting from this project.

**Acquisition of in vivo MRI data.** In vivo MRI data were acquired on an 11.7 Tesla BioSpec preclinical scanner running on Paravision 6.0 and equipped with a 72 mm diameter volume transmission coil and a 4-channel phased-array surface coil, optimized for rat brain, for reception (Bruker, Germany). Animals were anaesthetized using an air/$O_2$ mixture (50:50) and isoflurane (5% for induction, with maintenance concentrations varying from 0.75 to 2%, depending on the animal). They were then positioned in a dedicated cradle and placed within the magnet. During the functional acquisitions the breathing rate and body temperature were continuously recorded (using the SAM-PC software from SAII (SA Instruments, USA)). Body temperature was kept near 37 °C and breathing rate as close as possible to 70 bpm. The isoflurane concentration was adjusted for each animal to both control respiratory rate and thus limit as much as possible the potential influence of anesthesia on the animal's physiology. A Rapid Acquisition with Refocused Echoes (RARE) sequence with the following parameters was used to obtain a $T_2$-weighted anatomical image for each rat brain: RARE factor = 16, effective TE/TR = 33/1500 ms, field-of-view = 28.8 × 28.8 × 14.4 mm$^3$, acquisition matrix = 192 × 192 × 48, spatial resolution = 150 × 150 × 300 μm$^3$, 1 average, total acquisition time = 14 min 24 s. The resting-state functional MRI dataset (rs-fMRI) consisted of a series of $T_2$-weighted Blood-Oxygen-Level Dependent (BOLD) sensitive images, acquired using a multi-shot Spin-Echo Echo Planar Imaging (SE-EPI) sequence with the following parameters: 3 segments, TE/TR = 17.5/2000 ms, field-of-view = 24 × 24 × 20 mm$^3$, acquisition matrix = 64 × 64 × 20, spatial resolution = 375 × 375 × 750 μm$^3$, inter-slice gap = 0.25 mm, 450 repetitions, total acquisition time = 15 min. To reduce the artifacts resulting from movement associated with breathing, a trigger was used to ensure that functional images were only acquired during the most stable portion of the breathing curve. The trigger times were recorded by Paravision to be used during data post-processing. The rs-fMRI scan was performed at least twice, in two separate sessions, for each animal.

**Acquisition of ex vivo MRI data.** Ex vivo MRI data were acquired at the end of the in vivo experiments on 6 randomly chosen animals. Rats were euthanized with an overdose of sodium pentobarbital (100–150 mg.kg$^{-1}$, intraperitoneal injection), intracardiacally perfused with NaCl 0.9% to remove blood and then fixed with paraformaldehyde 4% in phosphate buffered saline (PBS, 0.01 M). After perfusion, animal heads were collected and post-fixed within paraformaldehyde 4% solution at 4 °C. 48 h prior to the MRI session, the paraformaldehyde solution was replaced by PBS (0.01 M) for tissue rehydration. To avoid any possible damage and deformation of the brains, they were kept inside the skulls during the entire procedure. For scanning, the head was placed into a custom-built MRI compatible tube filled with Fluorinert FC-40 (3 M, Belgium), a magnetic susceptibility-matching fluid. Ex vivo MRI acquisitions were performed on the same 11.7 Tesla BioSpec preclinical scanner as was used for the in vivo studies, using a 40 mm diameter transmit/receive volume coil (Bruker, Germany). The MRI protocol consisted of the acquisition of one $T_2^*$-weighted anatomical image using a Multiple Gradient Echo (MGE) sequence having the following parameters: TE/TR = 3/100 ms, 10 echoes spaced by 4 ms, field-of-view = 28.8 × 21.6 × 14.4 mm$^3$, acquisition matrix = 320 × 240 × 80, spatial resolution = 90 × 90 × 180 μm$^3$, 16 averages, total acquisition time = 8 h 32 min (Supplementary Fig. 1).

**Processing of anatomical MRI data.** All in vivo and ex vivo anatomical MRI data ($T_2$-weighted and $T_2^*$-weighted images) were reconstructed using homemade Matlab routines (MathWorks, USA, Matlab R2016b) and were saved to NIFTI format. For the ex vivo data, the sampled k-space was zero-filled[37,38] ultimately to finally reconstruct $T_2^*$-weighted images with a spatial resolution of 90 × 90 × 90 μm$^3$.

**Building of the ex vivo SIGMA anatomical brain template.** For each rat, we computed the sum of the 10 reconstructed $T_2^*$-weighted magnitude images (one per echo). Then, the sum images obtained for the 6 scanned animals were manually reoriented using the SPMmouse toolbox (http://www.spmmouse.org) of SPM8 software (Statistical Parametric Mapping, https://www.fil.ion.ucl.ac.uk/spm) and co-registered to the Tohoku brain template. Then, each co-registered image was segmented using the priors of the Tohoku brain template and the "New Segment" procedure of SPM8, to generate the corresponding probability maps of GM, WM and CSF using the Tohoku priors. This first set of TPMs, as well as the normalization parameters, were used in a second iteration to re-segment each brain, generating a new set of maps which were used in the DARTEL tool (Diffeomorphic Anatomical Registration Through Exponentiated Lie Algebra) to create our SIGMA template, aligned to the Tohoku and the Paxinos-Watson coordinate space. Through this approach we were able to recover the olfactory bulb as well as the hindbrain, both of which are missing in the original Tohoku brain template. In addition, the affine normalization maps obtained in the first step of the segmentation procedure were applied to the corresponding co-registered images and the normalized images were segmented again using the same procedure to obtain more accurate GM, WM and CSF priors. The 6 sets of priors were then used to generate the final template of GM, WM and CSF probability maps using the DARTEL approach for each tissue class[39]. The DARTEL algorithm also provides the flow field images containing the deformation parameters used to perform the inter-subject alignment of brain shapes[40]. The Jacobian image of each GM prior, derived

from the flow field image, was then applied to the corresponding normalized $T_2^*$-weighted image. Finally, the resulting 6 warped $T_2^*$-weighted images were averaged to create the final ex vivo SIGMA anatomical template of the rat brain (Supplementary Fig. 5).

**Building of the in vivo SIGMA anatomical brain template.** To build the in vivo anatomical template, we undertook a process similar to the one applied to the ex vivo MRI data. We selected images obtained from 32 rats, which met the criteria of exhibiting sufficient signal-to-noise ratio while demonstrating no ghosting artifact. After reconstruction of in vivo $T_2$-weighted images, all selected images were manually reoriented using the SPMmouse toolbox and co-registered to the ex vivo SIGMA brain template. Each co-registered image was then segmented as previously described, using the GM, WM and CSF probability maps of the ex vivo SIGMA template. After segmentation, the normalization parameters were applied to produce normalized $T_2$-weighted images that were segmented again to obtain more accurate GM, WM and CSF priors. Each set of priors was used to generate the final template of GM, WM and CSF probability maps using the same DARTEL approach, and the Jacobian image of each GM prior was applied to the corresponding normalized $T_2$-weighted image. During the last step, the resulting 32 warped $T_2$-weighted images were averaged to create the final in vivo anatomical template of the rat brain (Supplementary Fig. 6).

**Processing of functional MRI data.** All in vivo functional MRI data (EPI images) were reconstructed using homemade Matlab routines and saved in NIFTI format. After reconstruction, the trigger timing of respiratory rate was used to resample the data to a constant rate, in order to correct for the fact that breathing rhythm varied during the acquisition period. To minimize the amount of interpolation used while simultaneously avoiding loss of information, the new TR was set to the minimum across animals, which was found to be 2.3 s. This resampling procedure was followed by two pre-processing steps applied on FSL (FMRIB Software Library, https://fsl.fmrib.ox.ac.uk/fsl/fslwiki). These were: (i) slice timing correction with an odd order of slices; (ii) motion correction, using the mean EPI image as reference, and motion scrubbing for identification of outlier volumes in motion.

The mean SE-EPI image was then registered, using a linear registration with 7 degrees of freedom, to its corresponding anatomical $T_2$-weighted image and then normalized, using a combination of affine and non-linear registrations, to the in vivo anatomical brain template. The GM, WM and CSF tissue probability maps of the $T_2$-weighted images were summed to calculate a brain mask. This brain mask and the corresponding inverted brain mask were registered to the native space of EPI images and subsequently used to isolate both the brain tissue signal and the mean signal external to the brain, as well as to skull strip the EPI data. The 6 motion parameters, outlier volumes and global brain signals were regressed from the motion-corrected data and the residuals considered as signals of interest. The concatenation of all spatial transformations was applied to the residual data, in order to spatially normalize data onto the in vivo brain template. We followed this step by spatially smoothing the data using a Gaussian filter (1.5 voxels) and filtering the temporal signal using a band-pass filter between 0.005 and 0.1 Hz.

Finally, an Independent Component Analysis (ICA) was run for each subject to detect the remaining noise and artifact signals (Supplementary Fig. 2) using the MELODIC tool (Multivariate Exploratory Linear Optimized Decomposition into Independent Components) of FSL[41]. This process decomposes the fMRI signal into different time courses, which are in turn translated into spatial maps. The process thus quantifies the time-course association of each voxel, while optimizing the spatial statistical independence of the maps. The resulting components were visually inspected, and the ones identified as artefacts regressed from the original data[42].

**Definition of the ex vivo SIGMA anatomical brain atlas.** As previously mentioned, some rat brain atlases are available in MRI compatible format (Table 1). These atlases cover different parts of the rat brain with different levels of anatomical detail. Here we combined the Waxholm (sub-cortical emphasis) and the Tohoku (cortical emphasis) atlases over our ex vivo SIGMA anatomical template to create a new fully segmented rat brain atlas (Fig. 8a).

To do so, we used our high-resolution anatomical template as the reference and registered the template anatomical images associated with both atlases (Waxholm and Tohoku) to our template space. For the Waxholm atlas, an initial manual affine approximation was determined using the SPMmouse toolbox. This was followed by another linear registration calculated using the FLIRT command of FSL (FMRIB's Linear Image Registration Tool) with 12 degrees of freedom. As the Waxholm template is composed of a single very high-resolution acquisition, in addition to the great anatomical detail of the subject, it retained also the noise and artifacts from the acquisition. No non-linear registration was applied as it was found that these factors severally limited the performance of the algorithm, while the result of the linear registration was found to be satisfactory for the current purpose. For the Tohoku atlas, the FLIRT software of FSL was used to calculate a linear affine registration to our template with 12 degrees of freedom. This was followed by a non-linear registration performed with the FNIRT command of FSL (FMRIB's Nonlinear Image Registration Tool) using the previous affine registration as the initial approximation. The final registration transformations were applied to

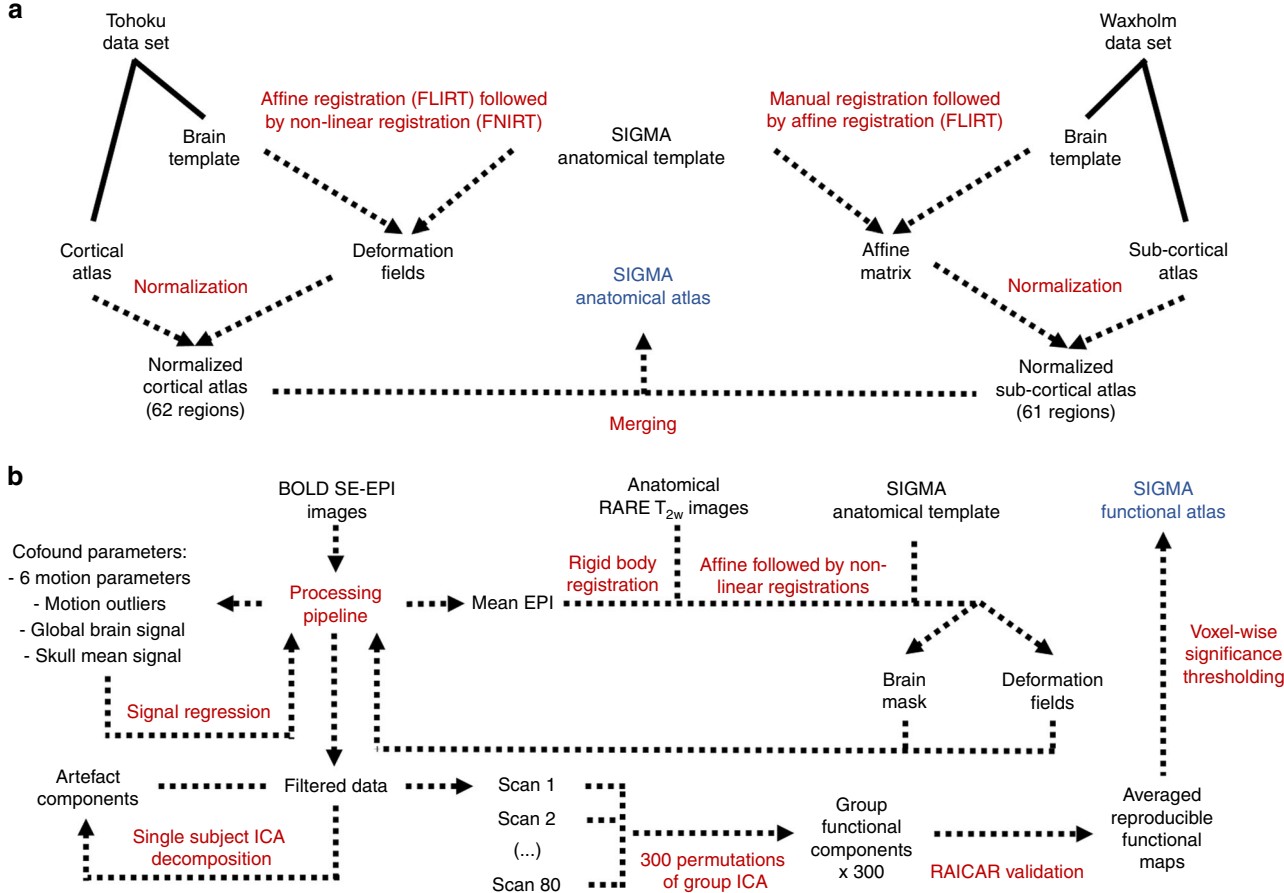

**Fig. 8 Schematic of the workflows developed to create SIGMA atlases of rat brain.** Workflows used for the creation of **a** the ex vivo SIGMA anatomical atlas of rat brain and **b** the in vivo SIGMA functional atlas.

the corresponding atlases using a nearest neighbor interpolation method to preserve the label intensities. All the registered labels of interest were then merged into a common image. We selected 61 normalized sub-cortical and brainstem structures from the Waxholm atlas and 62 normalized cortical structures from the Tohoku atlas. Of the registered areas, the matching of the Regions Of Interest (ROIs) corresponding to the WM tracts extending from the corpus callosum to the internal and external capsule was found to be less accurate than desired. To improve this specific segmentation, we used the WM tissue probability map to better isolate these tracts. The procedures consisted of thresholding the voxel values, identifying and isolating the cluster of voxels that corresponded to this area, thus producing a better delineation of the main WM fibber bundles in the brain. Finally, the left and right ROIs of the 123 selected structures from the Waxholm and Tohoku atlases were spatially merged to create the final ex vivo SIGMA anatomical atlas of the rat brain (Supplementary Table 1).

**Definition of the in vivo SIGMA functional brain atlas.** Following the processing of functional MRI data, a group ICA (gICA) was run on the 80 rs-fMRI scans (exhibiting sufficient signal-to-noise ratio and no excessive movement) using temporal concatenation (Fig. 8b). ICA was used to identify patterns of functional connectivity common to the different rats, analogous to what is known in humans as the resting state networks. Because this is a stochastic process, dependent on the conditions of initialization (the order by which the rs-fMRI acquisitions are introduced), solutions may represent a local optimal point and may not necessarily be reproducible.

To verify the validity of the maps found, we followed the general outline of the RAICAR methods (Ranking and Averaging Independent Component Analysis by Reproducibility)[27]. A number, $N$, permutations of the gICA were run by randomly mixing the order of the animals. The similarity between each map of each permutation was determined as the Pearson correlation coefficient. Next, the correlation between the maps and all others was calculated, creating $N$ matrices of spatial correlations. The pair of maps ($x$ and $y$ matrix point) with the highest correlation across all permutations was found. For each of the other permutations, the map with the highest correlation with $x$ was associated with this map. The process was repeated until no more maps across all permutations remained. To determine a reproducibility threshold, we calculated the logarithmic distribution (using 200 bins) of all the absolute values and identified its minimum, $m$. The

reproducibility, $R$, of each map was defined as the number of associated maps with a correlation value above $m$. Maps were considered reproducible when the $R$ value was equal or higher than half of the maximum value of $R$ found. The final maps were calculated as the average of all above-threshold associated maps. To determine the optimal $N$, we explored values between 5 and 400 in steps of five to determine the point where the $R$, the variance and the $m$ stabilize.

Using the averaged ICA maps with verified reproducibility, a functional atlas was built by thresholding each map to its top 3% voxels with the highest score. Voxels that survived this threshold in more than one map were assigned to the one with the higher $z$-score, forming a region of interest for each component. This size threshold ensured that each map would have enough voxels to be meaningful, as well meeting the requirement that each voxel exhibit a high correlation with the other associated voxels. Each ROI was restricted to a single cluster of voxels, except when two symmetric clusters were present: in that specific case, they were separated into two individual ROIs. Then, each ROI was visually inspected and labeled with the aid of the Paxinos-Watson atlas. Finally, ROIs were concatenated to create the final in vivo SIGMA functional atlas of the rat brain (Supplementary Table 3).

**Template comparison—Voxel-based morphometry (VBM) analysis.** In order to verify the viability of the SIGMA anatomical template and its corresponding tissue probability maps, we compared its use with the Tohoku resources in a VBM analysis. We compared the ability of both sets of resources to detect differences between a group of control ($n = 6$) and stressed animals ($n = 6$). Stressed animals were exposed to 21 days of a chronic unpredictable stress protocol, which we have previously shown in these animals to result in increased levels of corticosterone, anxious-like behavior and overall disruption of gray matter density and functional networks[26]. As the hippocampus is known to exhibit profound structural alterations in this paradigm[43–46], we focused the comparison on this structure, testing the ability of each resource to reveal change therein. Normalized in vivo $T_2$-weighted images of each animal were segmented into GM, WM, and CSF probability maps with SPM8 using either the Tohoku or the SIGMA priors. Then, each normalized GM image was warped using the deformation parameters calculated by the DARTEL algorithm and eventually modulated to correct volume changes occurring during the deformation step. The normalized-warped-modulated GM images were spatially smoothed with an isotropic Gaussian kernel (400 μm full

width at half maximum) to create Gray Matter Concentration maps (GMC). VBM analysis was performed using a two-sample Student *t*-test in SPM8 (control versus. stress). For each cluster, the significance of the peak voxel was set as $p < 0.05$. Results presented as surface results were obtained with the BrainNet Viewer[47] (https://www.nitrc.org/projects/bnv, Version 1.6), which generates the hippocampus mesh from anatomical atlas images. All statistical tests performed for the template comparison were two-sided tests.

**Atlas comparison—Functional connectivity analysis**. To further validate the use of the SIGMA anatomical and functional atlases, we also tested the properties of functional networks built using both atlases. Functional networks were obtained using the full rs-fMRI processed dataset ($n = 80$) by extracting the mean time-series of each ROI, using either the anatomical or functional atlas. We then calculated the Pearson correlation coefficient between each pair, creating for each subject a connectivity matrix. We described the degree distribution, small-worldness and modularity of both kinds of networks using the Brain Connectivity Toolbox (https://sites.google.com/site/bctnet)[31]. To understand if the use of ROIs having functional origin would represent more accurately the true signals, we compared the within-ROI signal variation at the voxel level to the ones obtained with the anatomical atlas using the non-parametric permutation testing randomize software of FSL. This analysis was restricted to voxels that belong to a ROI in both atlases. Results were considered significant at $p < 0.05$ (non-parametric permutation-based group comparison with a Student *t*-test), corrected for the Family Wise Error rate and using threshold-free cluster enhancement. In order to allow comparison of the two atlases, and specifically to address the unequal number of ROIs and spatial resolution, regions of the anatomical atlas were merged according to their description and systems, thereby creating a new anatomical atlas having 82 ROIs. All statistical tests performed for the atlas comparison were two-sided tests.

## Data availability

The datasets generated during and/or analyzed during the current study, including all the SIGMA resources, are available from the NITRC platform using the following link: https://www.nitrc.org/projects/sigma_template. The source data underlying Table 2, Figs. 3c and 7b, c, as well as Supplementary Fig. 3a–e, are provided as a Source Data file.

## Code availability

The code generated during the current study for the RAICAR analysis is fully available from the NITRC platform using the following link: https://nitrc.org/projects/sigma_template.

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

## Acknowledgements

This work is part of the SIGMA project with the reference FCT-ANR/NEU-OSD/0258/2012, co-financed by the French public funding agency ANR (Agence Nationale pour la Recherche, APP Blanc International II 2012), the Portuguese FCT (Fundação para a Ciência e Tecnologia) and the Portuguese North Regional Operational Program (ON.2—O Novo Norte) under the National Strategic Reference Framework (QREN), through the European Regional Development Fund (FEDER) as well as the Projecto Estratégico co-funded by FCT (PEst-C/SAU/LA0026-/2013) and the European Regional Development Fund COMPETE (FCOMP-01-0124-FEDER-037298). D.A.B. and A.N. were funded by grants from FCT-ANR/NEU-OSD/0258/2012. R.M. was supported by the FCT fellowship grant with the reference PDE/BDE/113604/2015 from the PhDiHES program. A.C. was supported by a grant from the foundation NRJ. P.M. was funded by Fundação Calouste Gulbenkian (Portugal; 'Better mental health during ageing based on temporal prediction of individual brain ageing trajectories TEMPO') with Grant Number P-139977. France Life Imaging is acknowledged for its support in funding the NeuroSpin platform of preclinical MRI scanners. The authors also acknowledge and thank Edward Ganz, MD, for proof reading our work.

## Author contributions

D.A.B. and R.M. acquired the MRI anatomical and functional data, developed the analysis pipelines with the assistance of P.M. and wrote the draft manuscript. A.N. participated to the in vivo animal experiments, including the induction of the rat model of chronic stress, whereas E.S. and F.G. performed the ex vivo experiments. F.B. and S.M. parametrized the MRI sequences and participated in the images acquisition and analysis. N.S. proposed the overall design of experiments for the SIGMA project supporting this work and P.M., F.M., J.C., J.C.S., M.B., T.M.J. and A.C. provided a critical and continuous feedback regarding the experimental design and the data analysis procedures. N.S., T.M.J., A.C., F.B. and S.M. managed the overall project and provided its funding. All authors contributed to the correction of the draft manuscript. D.A.B., R.M., N.S. and S.M. worked on the different revised versions.

## Competing interests

The authors declare no competing interests.
