## [Transparent Peer Review File · Nature Communications]

Reviewers' comments:

Reviewer #2 (Remarks to the Author):

In this article, the authors have embarked on an impressive effort to combine two MRI templates with corresponding structural atlas parcellations for the rat brain: the Waxholm and the Tohoku atlas. The motivation and technical challenges are well outlined in detail and the output is the registration of these two atlases into a new template: the SIGMA template. The authors created two versions of this new template: one ex vivo, an average of 6 warped ex vivo acquired T2*-weighted brain image volumes, and another in vivo, based on a larger group of animals subject to resting state functional MRI scans prior to the ex vivo scans. The in vivo template was warped onto the ex vivo template. The goal is referred to as developing a "comprehensive set of MRI references and resources, optimized for the rat brain, which would allow investigators to perform unified analyses of both structural and functional data". This is illustrated with a morphometric and connectivity analysis performed in this study.

Based on the clearly outlined goal, the new resources should be prepared for use of the broader scientific community. In a response letter to a previous review, it is stated that the resources described within the article will be fully available, under open access license. This needs to be firmed up in the manuscript and reviewed and verified. The present version of the manuscript does not contain this important information.

The paper provides detailed technical information on the acquisition and processing of the data, on the building the templates, etc. The methods used seem to be well suited for the purpose. A challenge is the extensive use of customized steps to create and merge the templates and atlases. In this regard, even if technical details are described, the study would be difficult to reproduce. For this reason, a comparison with the atlases of origin would be important to investigate and document. The illustrations provided in the present version of the manuscript are not suitable for this purpose.

The summary in the first paragraph of the Discussion indicates that the paper is not compatible with the goal stated in the Introduction. The authors should make up their mind. Is this primarily a technical paper (which is the message in the first paragraph of the Discussion, and certainly what the paper looks like) or a paper that the community can take as a starting point when trying use the new resources? One possible route would be not only to add the specific information on access to resources but also to add Supplementary information on a couple of use cases showing how the new resources would work and why they represent an improvement.

The Introduction refers to around 15 publications describing human brain atlases. Since so many papers are cited, it is tempting to look for the ones that are not, e.g. the Allen Human Brain Atlas (doi: 10.1038/nn.4171) or the Jubrain atlas (doi: 10.1016/j.neuron.2015.12.001). The authors should consider adding at least one reference to a review article or book chapter that would cover the more complete picture.

Reviewer #3 (Remarks to the Author):

This fine work proposes a comprehensive template for group analysis of rodent (rat) MRI data. The framework offered here is excellent, combines cleverly post-mortem and in-vivo data, as well as structural and functional (resting state) data, and thus can be of great usefulness for the community.

Except for aiming at the highest impact factor (typically as a result of internal or external pressure to do so), I cannot understand why the authors are seeking publication of such an obviously

technical work in Nature Communications, a journal dedicated to breakthroughs in discovery and knowledge. This work is better served by being published in one of the top journals of our community (Neuroimage for example), where its usefulness will be fully exploited and its visibility to its most relevant audience is the highest.

I will not repeat some of the justified concerns of reviewers of the previous incarnation of this work, e.g. the use of 2% isoflurane in conjunction with obtaining the resting state fMRI data, something that's obviously problematic and the reason why in such studies a combination of low level isoflurane and i.v. analgesic (alpha-chloralose) is typically used, and point out a couple of other points that should be considered when resubmitted elsewhere:

1. page 7 line 143: A TR of 1500ms in a T2-weighted introduces quite a bit of T1 weighting as well - why not longer TR?
2. The choice of T2-weighting and not T2* weighting for the resting state acquisition is not clear to me.
3. It has been asked by previous reviewers, but it's worth emphasizing that for template purposes, the choice of non-isovoxel acquisition is quite baffling.

Reviewers' comments:

Reviewer #2 (Remarks to the Author):

In this article, the authors have embarked on an impressive effort to combine two MRI templates with corresponding structural atlas parcellations for the rat brain: the Waxholm and the Tohoku atlas. The motivation and technical challenges are well outlined in detail and the output is the registration of these two atlases into a new template: the SIGMA template. The authors created two versions of this new template: one ex vivo, an average of 6 warped ex vivo acquired T2*-weighted brain image volumes, and another in vivo, based on a larger group of animals subject to resting state functional MRI scans prior to the ex vivo scans. The in vivo template was warped onto the ex vivo template. The goal is referred to as developing a “comprehensive set of MRI references and resources, optimized for the rat brain, which would allow investigators to perform unified analyses of both structural and functional data”. This is illustrated with a morphometric and connectivity analysis performed in this study.

1/ Based on the clearly outlined goal, the new resources should be prepared for use of the broader scientific community. In a response letter to a previous review, it is stated that the resources described within the article will be fully available, under open access license. This needs to be firmed up in the manuscript and reviewed and verified. The present version of the manuscript does not contain this important information.

As we have previously described and following the request of the reviewers, we have uploaded all the SIGMA resources to allow their complete evaluation by the reviewers, as well as to show our commitment to make these resources available. For this, we created the home page of the SIGMA rat brain atlas and template on the NITRC platform (https://www.nitrc.org/projects/sigma_template).

In the revised manuscript, the link has been added at the end of the Discussion section (lines 357-358).

2/ The paper provides detailed technical information on the acquisition and processing of the data, on the building the templates, etc. The methods used seem to be well suited for the purpose. A challenge is the extensive use of customized steps to create and merge the templates and atlases. In this regard, even if technical details are described, the study would be difficult to reproduce. For this reason, a comparison with the atlases of origin would be important to investigate and document. The

illustrations provided in the present version of the manuscript are not suitable for this purpose.

We agree with the reviewer that by the nature of the methods used, including several linear and non-linear registrations, they may be complex to reproduce directly. We have struggled with the very same question on how to better compare our merged atlas with the originals, but direct voxel-wise comparisons are impossible, as they do not overlap. Still, to try to address this point, we compared the volumes of each region of interest within the original atlases of Tohoku and Waxholm before and after the normalization procedure and documented this information in the manuscript. Briefly, we observed that the normalization procedure has slightly increased on average the volume of the regions of interest (+3.59%, Table S2). For some structures, the volume variations could be accentuated due to the Voronoi's diagram approach used to dilate competitively each region of interest, and therefore fill the space within the brain mask. On the other hand, some white matter structures were found to be shrunk (i.e. *fasciculus retroflexus*), likely because the mask used to delineate the white matter tracts was calculated from the average of 6 animals, instead of one in the Waxholm atlas. This comparison permits to have a complete overview of the effect of the normalization upon the brain atlases (Tohoku and Waxholm) used to create the SIGMA rat brain anatomical atlas.

These additional results have been added within the Results section (lines 158-166).

3/ The summary in the first paragraph of the Discussion indicates that the paper is not compatible with the goal stated in the Introduction. The authors should make up their mind. Is this primarily a technical paper (which is the message in the first paragraph of the Discussion, and certainly what the paper looks like) or a paper that the community can take as a starting point when trying use the new resources? One possible route would be not only to add the specific information on access to resources but also to add Supplementary information on a couple of use cases showing how the new resources would work and why they represent an improvement. The beginning of the discussion section has been rewritten in accordance with the introduction section, highlighting how our work should be considered as a starting point to unify the analyses of both structural and functional MRI preclinical data (lines 238-244).

The Introduction refers to around 15 publications describing human brain atlases. Since so many papers are cited, it is tempting to look for the ones that are not, e.g. the

Allen Human Brain Atlas (doi: 10.1038/nn.4171) or the Jubrain atlas (doi: 10.1016/j.neuron.2015.12.001). The authors should consider adding at least one reference to a review article or book chapter that would cover the more complete picture.

Following the reviewer's suggestions, we have added two references to this section, corresponding to review articles (line 71):

1. Mandal, P. K., Mahajan, R. & Dinov, I. D. Structural brain atlases: design, rationale, and applications in normal and pathological cohorts. *J. Alzheimers Dis. JAD* 31 Suppl 3, S169-188 (2012).
2. Cabezas, M., Oliver, A., Lladó, X., Freixenet, J. & Cuadra, M. B. A review of atlas-based segmentation for magnetic resonance brain images. *Comput. Methods Programs Biomed.* 104, e158-177 (2011).

Reviewer #3 (Remarks to the Author):

This fine work proposes a comprehensive template for group analysis of rodent (rat) MRI data. The framework offered here is excellent, combines cleverly post-mortem and in-vivo data, as well as structural and functional (resting state) data, and thus can be of great usefulness for the community.

Except for aiming at the highest impact factor (typically as a result of internal or external pressure to do so), I cannot understand why the authors are seeking publication of such an obviously technical work in Nature Communications, a journal dedicated to breakthroughs in discovery and knowledge. This work is better served by being published in one of the top journals of our community (Neuroimage for example), where its usefulness will be fully exploited and its visibility to its most relevant audience is the highest.

We understand the concerns of the reviewer and would like to assure that we have discussed the suitability of the journal in multiple occasions with the editors of the Nature group. Currently, our aim is to go beyond the traditional neuroimaging community and distribute our resources largely in the preclinical neuroscience community, which critically needs reference spaces and fully validated methods for the coregistration, normalization and mapping of MRI data. Sometimes, to investigate some specific brain functions or pathologies, the research team might not have the technical knowledge to build, find or adapt the resources that they need.

Furthermore, we strongly believe that the workflows, reference spaces and atlases developed here can be included within widely used platforms, such as SPM or FSL, allowing preclinical neuroscientists to establish their own data processing pipelines. Importantly, the SIGMA resources can become a standard in neurosciences to improve quality, comparability and reproducibility of preclinical MRI research, but also a critical tool to bridge the gap between preclinical and clinical studies to improve translational research in neurosciences.

I will not repeat some of the justified concerns of reviewers of the previous incarnation of this work, e.g. the use of 2% isoflurane in conjunction with obtaining the resting state fMRI data, something that's obviously problematic and the reason why in such studies a combination of low level isoflurane and i.v. analgesic (alpha-chloralose) is typically used, and point out a couple of other points that should be considered when resubmitted elsewhere:

Here, the reviewer raises an important point concerning the anaesthetic mixture used during a functional MRI acquisition, a question already raised within the first evaluation of our work.

Numerous studies and book chapters³⁻⁵ reported the effects induced by anaesthetics on the cerebral vasculature, but also on neurotransmitters concentration and release. From the systematic review proposed by Jennifer X. Haensel et al.³, we learnt that isoflurane is the most commonly used product to anaesthetize animals in imaging laboratories (for both induction and maintenance). Upon 126 eligible studies included in the review, isoflurane was used in 43.7% of them, halothane in 33,3%, α -chloralose in 7.1%, pentobarbital in 3.2 %, propofol in 1.6 % and chloral hydrate in one study (0.8 %). The reason of the popularity of isoflurane lies in its simplicity of use (no intravenous route), but also in its low toxicity, making it suitable for repeated anaesthesia procedures required by longitudinal studies. Nevertheless, the most popular does not necessarily imply the best one and it is correct to indicate that isoflurane obscures cortico-striatal functional connections, silences the subcortical activity and reduces the cerebral metabolic rate of oxygen (CMRO₂)⁶⁻⁹. As mentioned previously, we did an immense effort to minimize the confounding effect of isoflurane by adjusting its concentration during the functional MRI acquisition, in order to maintain the animal respiration rate between 70 and up to 80 breathes per minute; in this way, we aimed to preserve physiological functions and limit anaesthesia biases as much as possible for each animal.

Hence, to clarify our anaesthesia strategy, we previously wrote in lines 378-380: “Animals were anaesthetized using an air/O₂ mixture (50:50) and isoflurane (5% for induction, with maintenance concentrations varying from 0.75% to 2%, depending on the animal).” We also added in lines 384-386: “The isoflurane concentration was adjusted for each animal to both control respiratory rate and thus limit as much as possible the potential influence of anaesthesia on the animal’s physiology.”

In order to clearly identify the limitations due to the use of isoflurane, we added a paragraph at the end of the discussion section, in which we stress to the reader that the functional mapping described here could be biased by the anaesthesia protocol and why we decided to choose this approach (lines 340-349).

1. page 7 line 143: A TR of 1500ms in a T2-weighted introduces quite a bit of T1 weighting as well - why not longer TR?

We acquired our *in vivo* anatomical data using an 11.7 Tesla MRI preclinical scanner, where the contribution of the T₁-weighting is still limited using a TR of 1500ms.

Indeed, the T_1 is around 2073 ± 100.7 ms in gray matter and 1861.3 ± 73.5 ms in white matter at this intensity of magnetic field (11.7 T)¹⁰. By contrast, the T_2 in gray matter is around 30.7 ± 1 ms and 36.2 ± 1 ms in white matter at the same intensity of magnetic field¹⁰. As we used a TR of 1500 ms and an effective TE of 33 ms, the Gray/White matter contrast mainly comes from a T_2 -weighted component: starting from the MRI signal equation for a RARE sequence, the T_1 -weighted contrast between gray and white matters is estimated around 6.9 %, whereas the T_2 -weighted contrast is estimated around 15.1 %. Moreover, the justification of using a relatively short TR for the RARE anatomical acquisitions mostly relies on the compromise that we made to ensure a sufficient spatial resolution for defining an *in vivo* anatomical template ($150 \times 150 \times 300 \mu\text{m}^3$), while keeping a reasonable acquisition time (14 min 24s).

Like for the MGE imaging strategy implemented for creating the SIGMA *ex vivo* anatomical template, we computed the contrast-to-noise ratio between gray and white matter structures (CNR GM/WM) in the SIGMA *in vivo* anatomical template derived from our RARE imaging strategy:

SIGMA in vivo anatomical template	Gray Matter (GM)	White Matter (WM)
Number of voxels	368 811	196 206
Mean signal	32 734	28 828
Standard deviation	1 144	1 733
\Rightarrow CNR (GM/WM) for the SIGMA in vivo anatomical template = 1.88		

This CNR (GM/WM) value is quite similar to the one obtained for the Tohoku *in vivo* T_2 -weighted anatomical template ($\text{CNR}_{\text{Tohoku}} = 1.99$, see Table 2). Therefore, we confirm that our RARE imaging strategy guarantees in the derived SIGMA *in vivo* anatomical template a good compromise between spatial resolution and CNR between gray and white matter structures, which is useful for spatial registration and segmentation of MRI anatomical data.

2. The choice of T_2 -weighting and not T_2^* weighting for the resting state acquisition is not clear to me.

For *in vivo* resting-state functional MRI acquisitions (rs-fMRI), the main reason of choosing a T_2 -weighting strategy (SE-EPI, Spin-Echo Echo Planar Imaging), instead of a T_2^* -weighting strategy (GE-EPI, Gradient-Echo Echo Planar Imaging), is to achieve a good image quality, with a reduced sensitivity to artefacts caused by

magnetic susceptibility effects. These effects, induced by differences in magnetic susceptibilities (especially at the tissue/air interfaces), may cause geometric distortions, signal loss or brightening. Knowing that the impact of these undesirable artifacts is significantly increasing with the intensity of the static magnetic field, we decided to perform our rs-fMRI acquisitions at high magnetic field (11.7 Tesla) using a T₂-weighting strategy (SE-EPI).

3. It has been asked by previous reviewers, but it's worth emphasizing that for template purposes, the choice of non-isovoxel acquisition is quite baffling.

Regarding the *ex vivo* MRI anatomical data, the images were acquired using a 90 x 90 x 180 μm^3 spatial resolution firstly to keep a reasonable acquisition time, but also to reduce partial volume artefacts^{12,13}. To restore the isotropic resolution, we relied on the zero-filling strategy described by Bernstein MA *et al* (2001)¹². The zero-filling strategy is routinely used to expand the image matrix size in the phase-encoded or slice-encoded directions. In our 3D *ex vivo* imaging datasets, the zero-filling strategy did not add any information to the input raw data but improved the apparent spatial resolution of the reconstructed images in the slice direction while reducing the partial volume artefacts.

Regarding the *in vivo* MRI data, these datasets were acquired in the context of an extended project^{14,15}, where several sequences were performed on the same animal (anatomical and functional sequences, diffusion tensor imaging and spectroscopy protocol). Isotropic resolution acquisitions could have been performed but keeping a similar data quality would have required a proportional increase of the acquisition time. To ensure a reasonable acquisition time (critical for *in vivo* acquisitions), it is quite common in the MRI community to set the spatial resolution in the slice direction to twice the size of the in-plane voxel.

The references 37 and 38 have been added within the manuscript (line 421).

We thank again the two reviewers for their helpful feedbacks on our manuscript and the editorial board of *Nature Communications* for the interest in our work. Please note that in order to comply with the manuscript checklist of *Nature Communications*, we also modified the title of our manuscript, reduced the length of the abstract, moved the Materials and Methods section after the Discussion section and reordered the figures accordingly.

References

1. Mandal, P. K., Mahajan, R. & Dinov, I. D. Structural brain atlases: design, rationale, and applications in normal and pathological cohorts. *J. Alzheimers Dis. JAD* **31 Suppl 3**, S169-188 (2012).
2. Cabezas, M., Oliver, A., Lladó, X., Freixenet, J. & Cuadra, M. B. A review of atlas-based segmentation for magnetic resonance brain images. *Comput. Methods Programs Biomed.* **104**, e158-177 (2011).
3. Haensel, J. X., Spain, A. & Martin, C. A systematic review of physiological methods in rodent pharmacological MRI studies. *Psychopharmacology (Berl.)* **232**, 489 (2015).
4. Steward, C. A., Marsden, C. A., Prior, M. J. W., Morris, P. G. & Shah, Y. B. Methodological considerations in rat brain BOLD contrast pharmacological MRI. *Psychopharmacology (Berl.)* **180**, 687–704 (2005).
5. *Anesthesia and analgesia in laboratory animals.* (Elsevier [u.a.], 2008).
6. Kalthoff, D., Seehafer, J. U., Po, C., Wiedermann, D. & Hoehn, M. Functional connectivity in the rat at 11.7T: Impact of physiological noise in resting state fMRI. *Neuroimage* **54**, 2828–39 (2011).
7. Kalthoff, D., Po, C., Wiedermann, D. & Hoehn, M. Reliability and spatial specificity of rat brain sensorimotor functional connectivity networks are superior under sedation compared with general anesthesia. *NMR Biomed.* **26**, 638–650 (2013).
8. Liu, X., Zhu, X.-H., Zhang, Y. & Chen, W. The Change of Functional Connectivity Specificity in Rats Under Various Anesthesia Levels and its Neural Origin. *Brain Topogr.* **26**, 363–377 (2013).
9. Paasonen, J., Stenroos, P., Salo, R. A., Kiviniemi, V. & Gröhn, O. Functional connectivity under six anesthesia protocols and the awake condition in rat brain. *NeuroImage* **172**, 9–20 (2018).
10. Graaf, R. A. de. *In Vivo NMR Spectroscopy: Principles and Techniques.* (John Wiley & Sons, 2019).
11. Chavhan, G. B., Babyn, P. S., Thomas, B., Shroff, M. M. & Haacke, E. M. Principles, Techniques, and Applications of T2*-based MR Imaging and Its Special Applications. *Radiographics* **29**, 1433–1449 (2009).
12. Bernstein, M. A., Fain, S. B. & Riederer, S. J. Effect of windowing and zero-filled reconstruction of MRI data on spatial resolution and acquisition strategy. *J. Magn. Reson. Imaging JMRI* **14**, 270–280 (2001).
13. Du, Y. P., Parker, D. L., Davis, W. L. & Cao, G. Reduction of partial-volume artifacts with zero-filled interpolation in three-dimensional MR angiography. *J. Magn. Reson. Imaging JMRI* **4**, 733–741 (1994).
14. Magalhães, R. *et al.* The dynamics of stress: a longitudinal MRI study of rat brain structure and connectome. *Mol. Psychiatry* (2017). doi:10.1038/mp.2017.244
15. Magalhães, R. *et al.* A resting-state functional MR Imaging and Spectroscopy Study of the Dorsal Hippocampus in the Chronic Unpredictable Stress Rat Model. *J. Neurosci. Off. J. Soc. Neurosci.* (2019). doi:10.1523/JNEUROSCI.2192-18.2019

REVIEWERS' COMMENTS:

Reviewer #2 (Remarks to the Author):

The revised manuscript is clarifying, but the new resources still do not appear to be available to the reviewers. The authors have created a NITRC.org page for the new resource and provided a link in lines 357 - 358. So far, files are not available for downloading from NITRC.

[Comments on the online resources after these had been made available to the reviewers:]

NCOMMS-19-10436A Online resources

Problems:

1) Axes are non-standard, at least ITK Snap does not recognize them properly See ExVivo.png: (L)eft and (R)ight directions are presumably correct, but (S)uperior direction is swapped with (A)nterior and (I)nferior direction is swapped with (P)osterior As ITK Snap is considered a standard tool, and it is also free, it would be a nice touch to fix the axis directions in order to appear correctly in this software.

2) There are no label files, just Excel sheets (Labels.png). It is a safe bet that no anatomical toolkit is capable of using Excel sheets as labeling metadata (label names and colors)

3) Resolution mismatch with functional atlas. The Anatomical atlas has voxel resolution 260x184x342, which is identical to the resolution of ExVivo templates and thus they can be co-visualized in tools like ITK Snap. The Functional atlas has voxel resolution of 117x82x153, which obviously does not match the ExVivo template (nor the InVivo, 128x127x218), but it does not match the Functional template either, which has a radically different resolution, of 78x55x103 voxels.

It may be a different file from what is supposed to get shared, as SIGMA_Functional_Brain_Atlas.nii is the only NIFTI which has "spm - realigned" as description, all others contain "SIGMA_Wistar_Rat_Brain" in the description field.

Even if everything is as intended, it may be a nice addition to provide template(s) which have the same voxel resolution as the segmentation, certain software packages expect it.

Observations:

4) Masking differences

ExVivo brain image (ExVivo.png again) comes with surroundings (and a mask is provided), InVivo brain image (see InVivo.png) is masked already (but also comes with a mask). Is it intentional?

5) Providing separate volumes for Cerebrospinal Fluid, White Matter and Gray Matter looks very good.

Reviewer #3 (Remarks to the Author):

I appreciate very much the manner in which the authors responded to the critiques, those of mine and also by the other reviewers. I'm still in the opinion that the imaging community is better served by having this manuscript published in one of its "home journal", but having read the authors response, as well as the editor opinion on the matter, I'm recommending publication.

REVIEWERS' COMMENTS:

Reviewer #2 (Remarks to the Author):

The revised manuscript is clarifying, but the new resources still do not appear to be available to the reviewers. The authors have created a NITRC.org page for the new resource and provided a link in lines 357 - 358. So far, files are not available for downloading from NITRC.

We apologize to Reviewer #2 about the difficulties that she/he encountered when trying to download our SIGMA resources during the last reviewing step. The resources are now available for download on the NITRC platform using the following link: https://www.nitrc.org/projects/sigma_template.

[Comments on the online resources after these had been made available to the reviewers:]

NCOMMS-19-10436A Online resources

Problems:

1) Axes are non-standard, at least ITK Snap does not recognize them properly See ExVivo.png: (L)eft and (R)ight directions are presumably correct, but (S)uperior direction is swapped with (A)nterior and (I)nterior direction is swapped with (P)osterior As ITK Snap is considered a standard tool, and it is also free, it would be a nice touch to fix the axis directions in order to appear correctly in this software.

As highlighted by Reviewer #2, the orientation was swapped in the original SIGMA templates and atlases. According to the reviewer's comment, we used ITK-SNAP to correctly reorient the SIGMA brain templates and atlas (see **Figure 1** below).

Figure 1. Results of the overlapping between the reoriented brain anatomical template and the corresponding atlas using ITK-SNAP.

2) There are no label files, just Excel sheets (Labels.png). It is a safe bet that no anatomical toolkit is capable of using Excel sheets as labeling metadata (label names and colors).

According to the reviewer's comment, we have generated with ITK-SNAP two label files for both anatomical and functional atlases (see **Figure 2** below).

Figure 2. Label editor of ITK-SNAP fed with the anatomical atlas label file.

3) Resolution mismatch with functional atlas. The Anatomical atlas has voxel resolution 260x184x342, which is identical to the resolution of ExVivo templates and thus they can be co-visualized in tools like ITK Snap. The Functional atlas has voxel resolution of 117x82x153, which obviously does not match the ExVivo template (nor the InVivo, 128x127x218), but it does not match the Functional template either, which has a radically different resolution, of 78x55x103 voxels.

It may be a different file from what is supposed to get shared, as SIGMA_Functional_Brain_Atlas.nii is the only NIFTI which has "spm - realigned" as description, all others contain "SIGMA_Wistar_Rat_Brain" in the description field.

Even if everything is as intended, it may be a nice addition to provide template(s) which have the same voxel resolution as the segmentation, certain software packages expect it.

As highlighted by Reviewer #2, the images provided in the SIGMA resources were only aligned in the mm coordinate space, which is sufficient for FSL and SPM viewers but might not work for others. According to the reviewer's comment, we are now providing the functional atlas with three different voxel resolutions (206x184x342, 117x82x153 and 127x127x228) to extend its use to all neuroimaging software packages. Moreover, the description within the NIFTI header has been labelled as "SIGMA_Wistar_Rat_Brain_Functional_Atlas".

Observations:

4) Masking differences

ExVivo brain image (ExVivo.png again) comes with surroundings (and a mask is provided), InVivo brain image (see InVivo.png) is masked already (but also comes with a mask). Is it intentional?

It was an omission on our part. The unmasked images for both anatomical templates (*ex vivo* and *in vivo*) have been added to the SIGMA resources.

5) Providing separate volumes for Cerebrospinal Fluid, White Matter and Gray Matter looks very good.

We thank again Reviewer #2 for the very helpful feedbacks on our SIGMA resources and our manuscript.

Reviewer #3 (Remarks to the Author):

I appreciate very much the manner in which the authors responded to the critiques, those of mine and also by the other reviewers. I'm still in the opinion that the imaging community is better served by having this manuscript published in one of its "home journal", but having read the authors response, as well as the editor opinion on the matter, I'm recommending publication.

We thank again Reviewer #3 for the very helpful feedbacks on our manuscript.